# Pathogen transmission from vaccinated hosts can cause dose-dependent reduction in virulence

Richard I. Bailey[1]*, Hans H. Cheng[2], Margo Chase-Topping[1,3], Jody K. Mays[2], Osvaldo Anacleto[1¤], John R. Dunn[2‡], Andrea Doeschl-Wilson[1‡]

1 Division of Genetics and Genomics, The Roslin Institute, Easter Bush, Midlothian, United Kingdom,
2 USDA, Agricultural Research Service, US National Poultry Research Center, Avian Disease and Oncology Laboratory, East Lansing, Michigan, United States of America, 3 Usher Institute of Population Health Sciences & Informatics, University of Edinburgh, Edinburgh, United Kingdom

¤ Current address: Institute of Mathematical and Computer Sciences, University of São Paulo, São Carlos, Brazil
‡These authors are joint senior authors on this work.
* r.bailey@ed.ac.uk

**Data Availability Statement:** All datasets generated and analysed during the current study are available in the Edinburgh DataShare repository, https://doi.org/10.7488/ds/2725. The

## Abstract

Many livestock and human vaccines are leaky because they block symptoms but do not prevent infection or onward transmission. This leakiness is concerning because it increases vaccination coverage required to prevent disease spread and can promote evolution of increased pathogen virulence. Despite leakiness, vaccination may reduce pathogen load, affecting disease transmission dynamics. However, the impacts on post-transmission disease development and infectiousness in contact individuals are unknown. Here, we use transmission experiments involving Marek disease virus (MDV) in chickens to show that vaccination with a leaky vaccine substantially reduces viral load in both vaccinated individuals and unvaccinated contact individuals they infect. Consequently, contact birds are less likely to develop disease symptoms or die, show less severe symptoms, and shed less infectious virus themselves, when infected by vaccinated birds. These results highlight that even partial vaccination with a leaky vaccine can have unforeseen positive consequences in controlling the spread and symptoms of disease.

## Introduction

Vaccination is routinely used as an efficient and economical way to control the spread and symptoms of infectious diseases in humans and livestock. Vaccines vary in their protective properties [1,2], and although some completely block infection, others only prevent disease symptoms but not infection or onward transmission. The latter are termed 'leaky' or 'imperfect' vaccines. Leaky vaccines are commonly used to prevent or alleviate disease symptoms in livestock and are becoming more prevalent among human vaccines [3]. Leakiness allows pathogen populations to persist even at high levels of vaccination coverage [4], and reduced mortality of vaccinated individuals can lengthen their infectious period and hence promote the

citation for these data is: Dunn, John R; Cheng, Hans H; Doeschl-Wilson, Andrea; Bailey, Richard I; Chase-Topping, Margo; Mays, Jody; Anacleto, Osvaldo. (2019). Data for "Pathogen transmission from vaccinated hosts can cause dose-dependent reduction in virulence", 2017-2018 [dataset]. University of Edinburgh. Roslin Institute. https://doi.org/10.7488/ds/2725.

**Funding:** This project was supported by Agriculture and Food Research Initiative Competitive Grant no. 2016-67015-24914 from the USDA National Institute of Food and Agriculture (all authors). RIB's and ADW's contributions were partly funded by the BBSRC Institute Strategic Funding Grant to The Roslin Institute (BBS/E/D/20002173). The funders had no role in study design, data collection and analysis, decision to publish, or preparation of the manuscript (all authors).

**Competing interests:** The authors declare no competing interests.

**Abbreviations:** DPC, days post-contact; DPI, days post-infection; FVL, feather viral load; GAPDH, Glyceraldehyde 3-phosphate dehydrogenase gene; HVT, herpesvirus of turkeys (live virus vaccine); IA, intra-abdominal; MD, Marek disease; MDV, Marek disease virus; NMS, nonmetric multidimensional scaling; PBL, peripheral blood lymphocyte; PBS, phosphate buffer solution; qPCR, quantitative polymerase chain reaction; vMDV, virulent MDV.

evolution of increased pathogen virulence [5]. A better understanding of the overall impacts on populations of vaccination with leaky vaccines is therefore urgently needed.

The underlying hypothesis in this paper is that vaccination with leaky vaccines not only has direct positive effects on vaccinated individuals but also indirect positive effects on individuals in the same contact group. Often only a fraction of a population receives the direct benefits of vaccination because of incomplete coverage and heterogeneity in vaccine responses [6–8]. However, vaccination even with a leaky vaccine often reduces pathogen load in infected individuals [5,9–15], with potential consequent reduction in the exposure dose of susceptible individuals. Transmission experiments, in which infected 'shedders' are placed in contact with uninfected 'contact' individuals and transmission recorded, have revealed that lower shedder pathogen load reduces transmission in some cases [5,16,17] but not all [13]. Measures of vaccine effectiveness can include these indirect benefits for unvaccinated individuals through dose-dependent reduction in transmission rates from infected vaccinated individuals [18]. However, beyond transmission effects, lower exposure dose can also decrease pathogen load in newly infected hosts [19–21], potentially leading to decreased pathogen virulence [19–20,22–27] and infectiousness in these secondary cases. These downstream effects of leaky vaccines on disease development and spread are currently poorly understood. Here, we use transmission experiments with vaccinated versus sham-vaccinated shedders and only unvaccinated contact individuals to examine how a leaky vaccine affects both transmission and subsequent pathogen virulence and load (and hence, potentially, infectiousness) in contacts.

*Gallid alphaherpesvirus* 2, more commonly referred to as Marek disease virus (MDV), is a highly oncogenic herpesvirus of poultry causing worldwide annual losses of US$1 to US$2 billion [28]. It is an airborne pathogen, spreading via inhalation of virus-laden 'chicken dust', which accumulates through shedding of infectious feather follicle epithelia [29]. Marek disease (MD) symptoms include peripheral nerve enlargement, tumours in a variety of organs, wing and leg paralysis, and iris lymphoma causing pupil irregularities, as well as death. Infected birds remain infectious for life, and the virus can remain infectious in the environment for many months. Higher MDV ingestion dose has been reported to increase disease progression [27,30], but this effect has not previously been linked to vaccination or exposure dose under natural transmission. On top of clear welfare concerns, MD causes production losses at inspection because of a drop in egg production of laying hens [31] and symptoms known as 'leukosis', leading to meat condemnation. Leukosis has other causative agents but is primarily due to MDV in chickens [32].

Management of MD led to development of the first widely used anti-cancer vaccine, the related live turkey herpesvirus *Meleagrid alphaherpesvirus 1*, commonly referred to as herpesvirus of turkeys (HVT) [33]. In the United States, vaccination of all commercial poultry has been routine since the 1970s. However, from the 1950s to the present day, there have been several jumps in MDV virulence [34], each causing more severe symptoms and reducing the symptom-blocking effects of existing vaccines. Several generations of vaccines have been developed to combat this increased virulence, all of which are leaky and may in fact have contributed to continuing virulence evolution [5]. Currently, widespread vaccination leads to low production losses in the US commercial poultry industry. However, vaccination is not routine worldwide and may vary considerably in quantity and quality [35], leading to incomplete effective vaccine coverage within a flock.

All MD vaccines including HVT are modified live viruses and are therefore potentially transmissible. Whenever transmissible live vaccines are used, vaccine transmission itself can potentially confer some secondary downstream protection in unvaccinated contacts, in addition to the effect of reduction in transmission of pathogenic virus. The more recently developed and widely used CVI988 (Rispens) MD vaccine is highly transmissible [36]. However,

despite quite extensive shedding of HVT vaccine virus into the environment [37,38], HVT transmission is usually low and is thought to be absent from young birds <8 weeks old vaccinated with low doses [39–41].

High variability in virulence among MDV strains [42], in genetic resistance among chicken strains [43], and in vaccine effectiveness [44] and transmissibility, combined with well-developed empirical methods for examining host infection and disease [45], make MDV in chickens an ideal model system to examine the relationships between vaccination with leaky vaccines and pathogen load, transmission, and subsequent virulence in unvaccinated birds.

The overall aim of this study was to assess how vaccination with a leaky vaccine affects pathogen transmission and subsequent disease development in unvaccinated contact individuals. To investigate this, we carried out transmission experiments for MDV in chickens, in which HVT-vaccinated or sham-vaccinated shedder birds inoculated with a virulent (vMDV) pathogen strain were placed in contact with unvaccinated naïve contact birds (Fig 1). We chose HVT vaccine because of its low transmissibility, its wide usage both to combat MDV and as a vector vaccine, and our extensive previous experience with this vaccine allowing optimization of experimental methods. We chose to focus solely on a well-studied vMDV (rather than more virulent vvMDV or vv+MDV) pathogen strain to allow comparison with many past studies and to maximize replication and hence our ability to detect differences in downstream effects. We used unvaccinated contacts to avoid confounding effects of vaccination on contact bird resistance and shedder transmission ability. We investigated to what extent vaccination reduces both MDV transmission and subsequent disease severity in contacts and asked whether the effects of shedder vaccination on contacts were mediated by lower shedder viral load. We found that shedder vaccination led to a large reduction in contact bird disease symptoms and provide strong evidence that this effect was mediated by pathogen load.

## Results

### Establishing the transmission model and sampling times

Unless otherwise stated, 'transmission', 'virus', and 'viral load' refer to the pathogenic MDV strain and not the vaccine virus strain. Appropriate contact duration and sampling times to examine shedder vaccination effects needed to be established in pilot experiments prior to the main trial (S1 Text). As little as 4 hours of contact between inoculated shedders and uninfected contacts was sufficient for most contact birds to become infected and show visible disease symptoms by 8 weeks post-contact (S2 Text, S1 Fig). A contact duration of 2 days was subsequently chosen to ensure ample shedding time and to standardize time available for shedding of feather follicle epithelia by the shedders. Both vaccinated and sham-vaccinated shedders were positive for small quantities of virus in feather follicle epithelia by 7 days post-infection (DPI), but this feather viral load (FVL) had increased considerably by 10 to 12 DPI (S3 Text, S2 Fig). When shedders were moved to a new set of contacts every 2 days from 10 to 20 DPI, the proportion of infected contacts, as measured by quantitative polymerase chain reaction (qPCR) from feather and blood samples collected 14 days post-contact (DPC), was consistently high (S3 Text, S3 Fig). However, although contact with sham-vaccinated shedders also consistently led to high incidence of disease symptoms at necropsy, contact with vaccinated shedders led to lower proportion of diseased contacts, in particular at the early contact periods. These temporal trends coincided with differences in shedder FVL, with higher overall FVL in sham-vaccinated birds, reaching a peak around 12 DPI and lower FVL peaking around 20 DPI in vaccinated shedders (S2 Fig). Both groups of shedders then remained positive for virus in feathers for the 8-week duration.

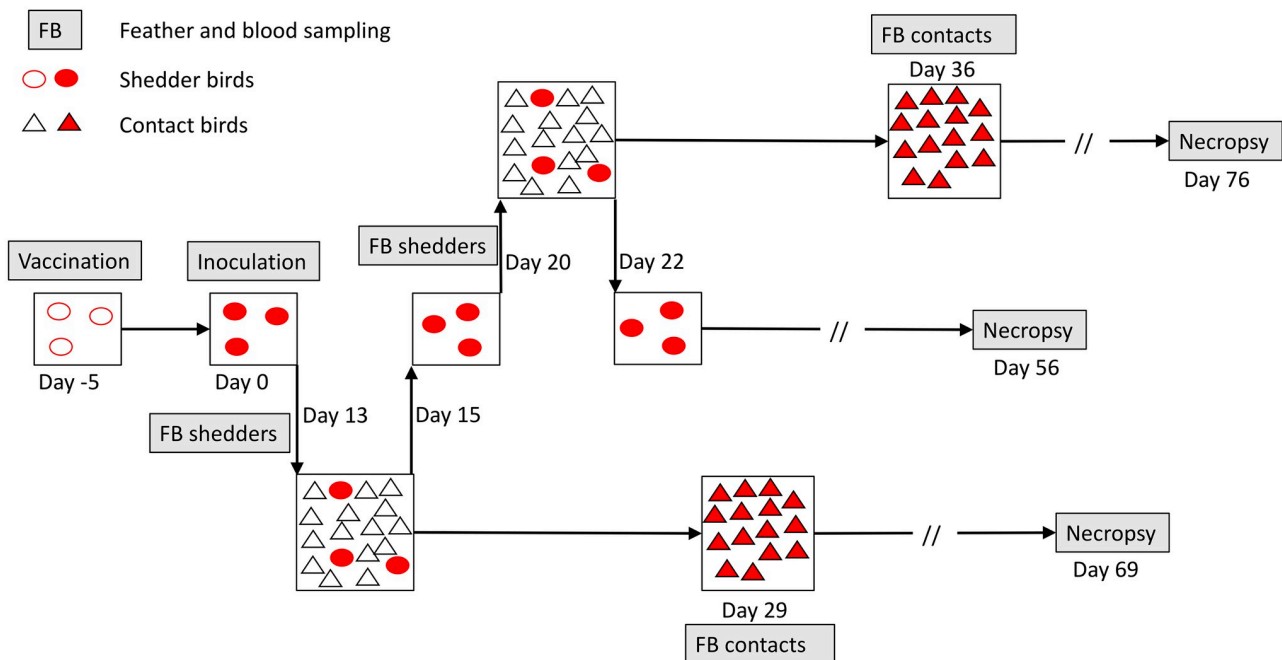

**Fig 1. Schematic overview of one 'lot' (2 lots per replicate, 1 for each shedder vaccination status) of the MD transmission experiment.** In each lot, shedder birds were all either HVT-vaccinated or PBS sham-vaccinated. All contacts were unvaccinated. The experiment comprised 16 replicates, each consisting of 1 lot in which 3 infected vaccinated shedders were placed in 48-hour contact with 15 naïve unvaccinated contacts at 2 time points, and 1 equivalent lot with sham-vaccinated shedders (4 additional sham-vaccinated lots were added because 2 of these had only 2 shedders because of early death). In total, there were 1,080 contacts and 106 shedder individuals distributed into 72 contact bird groups. Contact bird groups each had roughly equal numbers of males and females. All indicated time points (not to scale) are relative to the day of shedder inoculation with wild-type virus. Open and closed symbols represent uninfected and infected chickens, respectively. For all birds, necropsy was carried out to determine the presence and severity of disease symptoms (tumours and peripheral nerve enlargement) at 8 weeks post-infection (shedders) or post-contact (contact birds), or upon death/euthanasia, whichever was the sooner. HVT, herpesvirus of turkeys; MD, Marek disease; PBS, phosphate buffer solution.

Informed by the pilot data, we chose 13 and 20 shedder DPI as standardized contact start times in the main experiments (Fig 1) to capture the aforementioned temporal variation in vaccine effects and a 2-day contact period between shedders and contacts. Fourteen DPC was chosen as the time for contact bird blood and feather sampling, because it proved ample for build-up of FVL in infected contacts while minimizing among-contact transmission (S4 Fig). Viral loads were highly correlated between blood and feathers (main experiment contact birds only, correlation coefficient r = 0.73) and were typically higher and more often above the detection threshold in feathers, as shown previously [46]. Hence, we focused on viral load in feathers for all analyses because of the dual benefits of the typically above-threshold level viral loads and the fact that feathers are the infectious tissue, hence increasing the likely association with infectiousness. Examination of the presence and severity of disease symptoms (tumours and peripheral nerve enlargement) at necropsy took place at 8 weeks post-infection (shedders) or post-contact (contacts), or when moribund, if this occurred earlier. The subsequent results only refer to analyses of data from the main experiment illustrated in Fig 1.

## Vaccination blocks shedder disease symptoms without blocking infection

As expected, all shedder birds were positive for MDV as determined by qPCR, and vaccination almost universally blocked the development of disease symptoms at necropsy. Eighty out of eighty-six sham-vaccinated shedders (93%; 4 out of 90 birds excluded because of early death

from other causes) were MD-positive at necropsy, whereas only 5 out of 80 (6%) vaccinated shedders were MD positive.

## Shedder vaccination does not block transmission but reduces contact bird disease development and pathogen load

The complete set of contact bird analysis results are presented in S1 Table. Overall, vaccination of shedders did not block virus transmission but dramatically reduced the negative impacts of infection in contact birds. Almost all contacts became infected regardless of shedder vaccination status or DPI, with 100% (all 572 birds) contact bird infection for sham-vaccinated shedders and 97.4% for vaccinated (442 out of 454). This difference, albeit small, was significant, with contacts of sham-vaccinated shedders 0 to 0.28 times as likely to remain uninfected as contacts of vaccinated (Fisher exact test: 16.82, $p < 0.001$, odds ratio = 0, 95% CI 0–0.28). However, fewer infected contacts of vaccinated shedders developed visible disease symptoms or died within 8 weeks (Fig 2A), and of those showing visible symptoms, shedder vaccination was associated with less severe contact bird symptoms, including fewer tissues with tumours and less severe enlargement of peripheral nerves, as illustrated by nonmetric multidimensional scaling (Fig 2B).

Infected contacts were much less likely to show visible disease symptoms at necropsy after contact with vaccinated (232 out of 437 contacts; 53%) than sham-vaccinated (558 out of 569; 98%) shedders (Table 1). Disease symptoms in infected contacts were also more likely in the 20 DPI than 13 DPI contact groups (mixed-model logistic regression: $z = 4.5$, $p < 0.0001$), but this temporal effect was smaller when shedders were sham-vaccinated (vaccination status by DPI interaction; $z = -2.3$, $p < 0.05$). Males were marginally less likely to show visible disease symptoms than females ($z = -1.9$, $p = 0.05$).

Mortality rates were also much lower among infected contacts exposed to vaccinated shedders (Fig 3), with those exposed to sham-vaccinated shedders being 6 times more likely to die per unit time (95% CI 3.9–8.4; Table 1). Controlling for vaccination effects, contacts exposed to shedders at 20 DPI were almost twice as likely to die as those exposed to shedders at 13 DPI (95% CI 1.3–2.1; $z = 3.6$, $p < 0.0005$).

Among contacts positive for disease symptoms at necropsy, shedder vaccination led to significantly lower disease severity (number of tissues with tumours and enlargement of 3 peripheral nerves; see Fig 2B) for all individual symptoms except vagus nerve enlargement (Table 1). There was no evidence for an increase in contact bird disease severity between shedder DPI 13 and 20 for either vagus nerve (mixed-model ordinal logistic regression: $z = -0.1$, $p = 0.89$) or tumours ($z = 1.2$, $p = 0.21$) but marginal evidence for greater brachial nerve enlargement ($z = 1.9$, $p = 0.06$) and a significant increase in sciatic nerve enlargement ($z = 3.1$, $p < 0.005$) associated with the later exposure time. Regardless of the shedder vaccination status and exposure time, disease severity was significantly higher in contact females than males for all symptoms (Fig 2B; S1 Table).

Next, we tested the extent to which shedder vaccination status also influenced contact FVL as an indicator of the infectiousness of contact birds, which has potentially important knock-on effects for epidemiological dynamics. Infectiousness is likely to be determined by the amount of virus shed into the environment. Across all individuals, contact bird FVL at 14 DPC was much higher when exposed to sham-vaccinated than vaccinated shedders (Table 1, Fig 4). Contact FVL was also higher when exposed to shedders at 20 DPI than 13 DPI (mixed-model linear regression: $t = 4.9$, $p < 0.0001$).

In summary, contact birds exposed to vaccinated shedders still became infected but were considerably less likely to develop disease, experienced milder symptoms and lower mortality, and had lower FVLs.

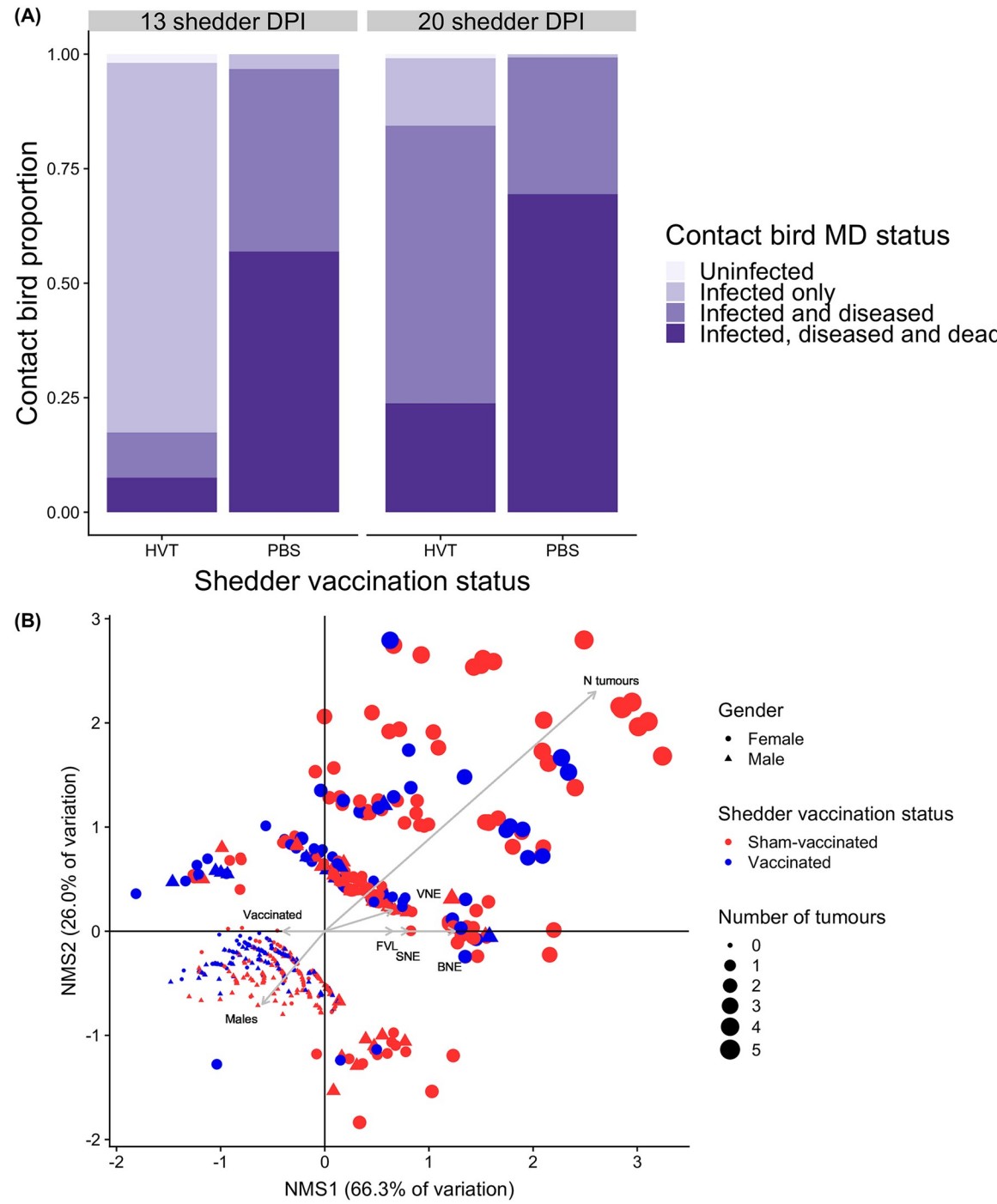

**Fig 2. Summary of shedder impacts on contact birds.** (A) Impact of shedder vaccination status on contacts at 13 and 20 shedder DPI. Contacts positive for virus in qPCR from samples taken at 14 DPC were classified as infected. 'Diseased' individuals showed visible symptoms (peripheral nerve enlargement and/or tumours) at necropsy, 8 weeks post-contact, or upon death. 'Dead' contacts were those that died or were humanely euthanized before the end of the 8-week experimental period, were infected, and were positive for disease symptoms at necropsy. HVT = vaccinated shedders; PBS = sham-vaccinated shedders. (B) Nonmetric multidimensional scaling plot, for diseased contacts only, of relationships between contact bird disease severity variables and contact bird sex, shedder vaccination status, and shedder FVL. Points are individual contact birds. Grey arrow distance along each axis represents its nonparametric Kendall's tau correlation with that axis. Opposite-pointing arrows indicate negative associations. Contacts of vaccinated shedders and males therefore had fewer tumours and less extreme nerve enlargement. Points are clustered from bottom-left to top-right into increasing numbers of tissues with tumours, concordant with changing point size; clustering in other directions indicates qualitatively different combinations of tissues with tumours. Variables differ qualitatively (binary, continuous, or ordinal) and so correlation coefficients and hence arrow

lengths may not be directly comparable. Shedder DPI effects not shown (see Results text and S1 Table). Underlying data are in Edinburgh DataShare repository (https://doi.org/10.7488/ds/2725). BNE, brachial nerve enlargement; DPC, days post-contact; DPI, days post-infection; FVL, feather viral load; HVT, herpesvirus of turkeys; MD, Marek disease; NMS1, nonmetric multidimensional scaling axis 1; qPCR, quantitative polymerase chain reaction; SNE, sciatic nerve enlargement; VNE, vagus nerve enlargement.

## Shedder vaccination effects on contacts are mediated by FVL

We hypothesized that the effects of shedder vaccination on contacts, described above, were mediated by a reduction in shedder FVL with vaccination, leading to a reduction in contact exposure dose with knock-on effects for disease development. To test this hypothesis, we followed the protocol for process analysis using regression, outlined in the Statistical Analysis section of Methods and materials. Before this, we tested whether HVT vaccine transmission occurred and might contribute to the described downstream effects.

HVT-specific qPCR on peripheral blood lymphocyte (PBL) samples of all contact birds from 6 contact bird groups (3 groups with 0 contact bird mortality and 3 with high mortality) revealed that only 8/89 (9%) unvaccinated contact birds were positive for HVT. HVT-positive birds were evenly distributed across contact groups, with 5/6 groups containing at least 1 positive bird (1 low-survival group had no positive birds), and no group containing more than 2 positive birds. According to Fisher exact tests, there were no significant differences in proportions positive for HVT between high- and low-survival groups (Fisher exact test: $p = 0.71$, odds ratio = 0.59, 95% CI 0.09–3.26). HVT transmission was unexpected given the young age of shedders and low vaccination dose [39–41] but was nevertheless too low to provide statistically significant evidence for secondary protective effects impacting contact bird FVL and disease progression. We therefore did not explicitly consider HVT vaccine transmission in our subsequent analyses, while acknowledging the possibility that transmission of undetectably small quantities of HVT that may enhance the downstream effects of vaccinated shedders may exist.

**Table 1. Effects of shedder vaccination status on contact bird disease symptoms, mortality and FVL for a model also including contact bird sex and shedder DPI, but excluding DPI by vaccination status interaction.** Full results, including models with the interaction, are in S1 Table.

| Contact bird response | Shedder vaccination coefficient (SE)[6] | Test statistic[7] | *p*-value |
|---|---|---|---|
| Disease status[1] | 8.19 (1.50) | 5.45 | <0.0001 *** |
| Mortality[2] | 1.74 (0.20) | 8.76 | <0.0001 *** |
| *N* tissues with tumours[3] | 0.50 (0.13) | 3.71 | <0.0005 *** |
| Vagus nerve enlargement[4] | 0.22 (0.23) | 0.94 | 0.35 |
| Brachial nerve enlargement[4] | 1.30 (0.26) | 5.01 | <0.0001 *** |
| Sciatic nerve enlargement[4] | 1.30 (0.24) | 5.36 | <0.0001 *** |
| FVL[5] | 1.98 (0.11) | 18.3 | <0.0001 *** |

[1]Infected contacts (qPCR) only. Logistic regression. Coefficient = mean log odds ratio for presence of contact disease symptoms when exposed to sham-vaccinated relative to vaccinated shedders.

[2]Infected contacts (qPCR) only. Cox proportional hazards. Coefficient = log hazard ratio of contact death at a given time point associated with sham-vaccinated relative to vaccinated shedders.

[3]Diseased contacts (necropsy) only. Poisson GLM. Coefficient = mean difference in number of contact tissues containing tumours with sham-vaccination.

[4]Diseased contacts (necropsy) only. Ordinal logistic regression. Coefficient = proportional log odds of an increase in contact nerve enlargement ranking with sham-vaccination.

[5]All contacts. Least square mean difference in contact bird $\log_{10}$(viral load + $1 \times 10^{-5}$) with sham-vaccinated relative to vaccinated shedders.

[6]Positive = increase in contacts when exposed to sham-vaccinated relative to vaccinated shedders, except for FVL (>1 = increase with sham-vaccination).

[7]t statistic for linear regression, z statistic for all other models.

**Abbreviations:** DPI, days post-infection; FVL, feather viral load; GLM, generalized linear model; qPCR, quantitative polymerase chain reaction; SE, standard error.

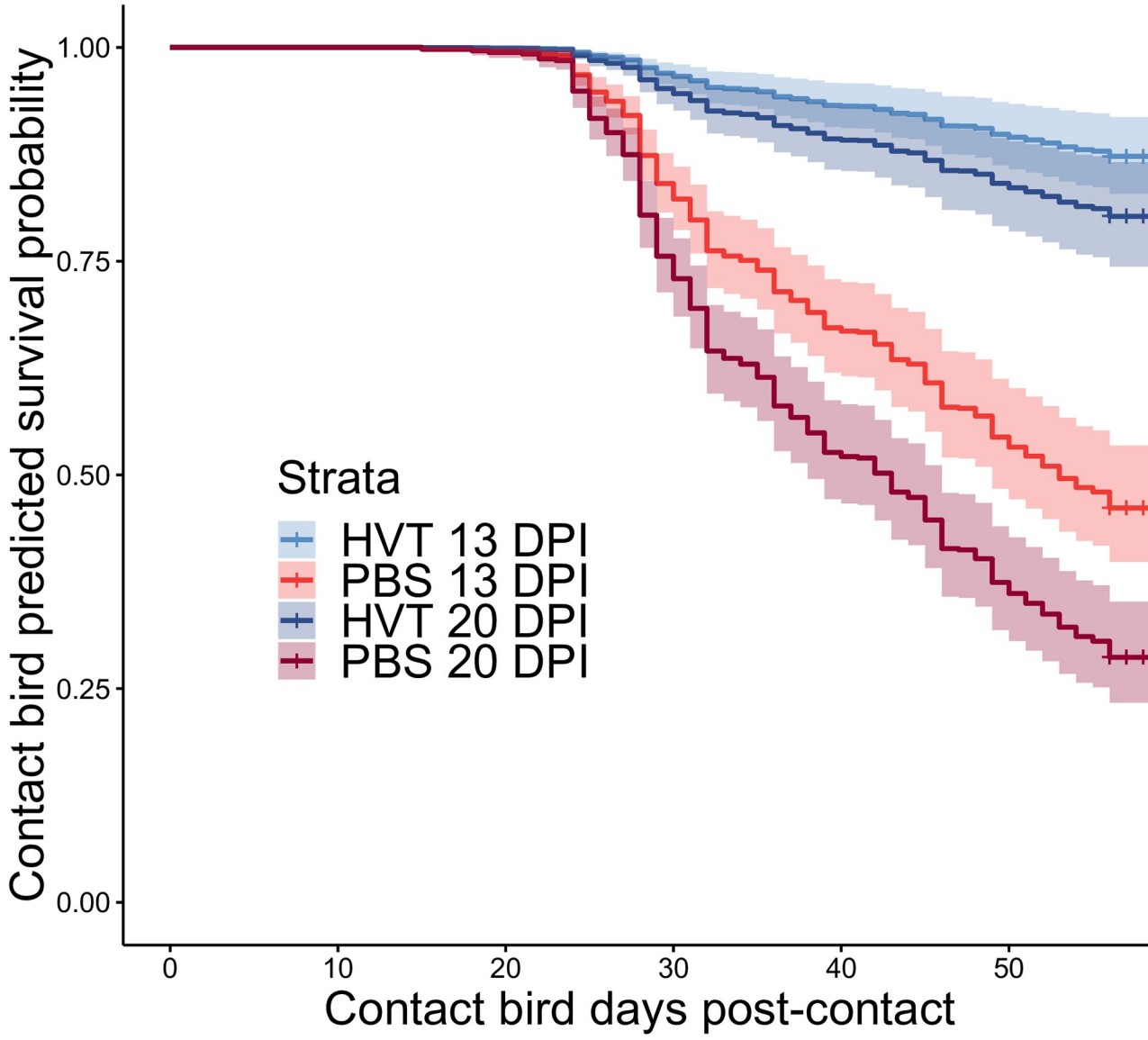

**Fig 3. Cox proportional hazards estimated survival probability curves for all combinations of shedder vaccination status and DPI.** Shaded areas represent 95% confidence intervals. HVT = vaccinated shedders; PBS = sham-vaccinated shedders. Note that all mortality up to 7 DPC was assumed to be chick mortality unrelated to MD, and these individuals were excluded from all analyses. Underlying data are in Edinburgh DataShare repository (https://doi.org/10.7488/ds/2725). DPI, days post-infection; DPC, days post-contact; HVT, herpesvirus of turkeys; MD, Marek disease.

Sham-vaccinated shedders had much higher FVL than vaccinated (mixed-model linear regression: t = 13.35, $p < 0.0001$; Fig 4). There was a highly significant increase in shedder FVL at 20 DPI over 13 DPI (t = 7.49, $p < 0.0001$), but the highly significant interaction between vaccination status and DPI (t = −5.03, $p < 0.0001$) revealed that this temporal change only occurred in vaccinated shedders.

Replacing shedder vaccination status with shedder FVL as a covariate in the statistical models for contact birds revealed that the effects of shedder FVL on contacts followed the same pattern as the effects of shedder vaccination status. Higher shedder FVL led to a small but significant increase in contact bird infection probability (univariate logistic regression: log odds = 0.76, z = 3.0, $p < 0.005$), with predicted infection probability increasing from 0.946 at

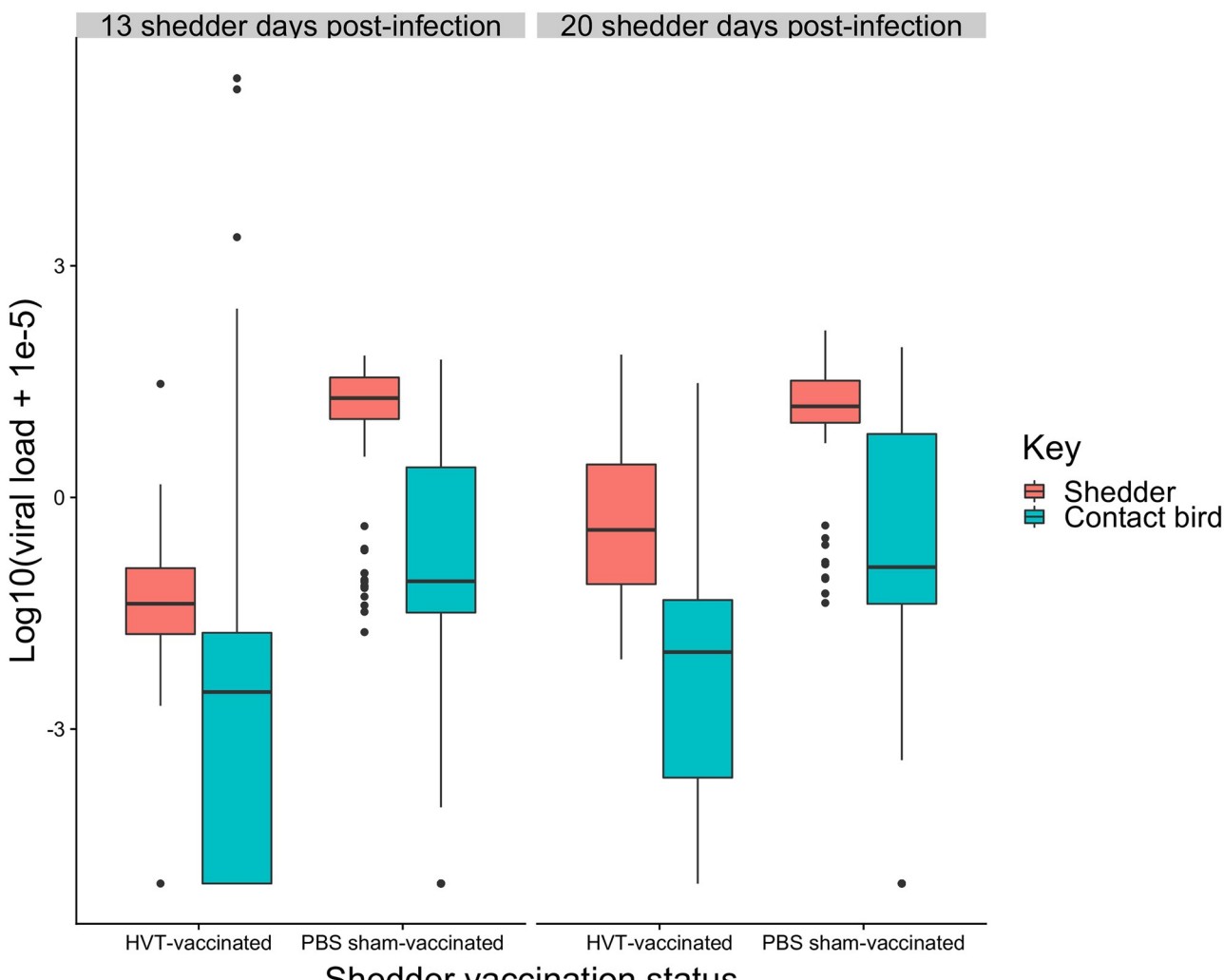

**Fig 4. Box and whisker plot of shedder and contact FVLs at 13 and 20 DPI.** Horizontal bars are medians. Boxes cover the first to third quartile, and vertical lines extend to maxima and minima except in the presence of outliers (filled circles). Shedder feather samples were taken at the start of each contact period, contact samples at 14 DPC. A small value ($1 \times 10^{-5}$) was added to FVL values prior to log transformation, and hence the $\log_{10}$(viral load) for birds negative for virus was −5. Underlying data are in Edinburgh DataShare repository (https://doi.org/10.7488/ds/2725). DPC, days post-contact; DPI, days post-infection; FVL, feather viral load.

the lowest shedder FVL values to 0.997 at the highest. Higher shedder FVL led to greater contact bird disease prevalence and severity, greater mortality, and higher contact FVL (Table 2).

Including shedder FVL in a model alongside vaccination status reduced but did not always remove the significance of vaccination status for all contact bird disease variables (S1 Table). This indicated that shedder FVL at least partially explained the impacts of shedder vaccination on infected contacts. However, the further addition of contact FVL and sum of contact group-mate FVL (the latter to account for possible among-contact infection during the 8-week experimental period) as predictors fully explained the effects of shedder vaccination on contact disease and survival, rendering shedder vaccination status nonsignificant in all models (Fig 5). The results indicate that shedder vaccination effects on contacts are fully mediated by FVL of shedders and infected contacts.

**Table 2. Effects of shedder FVL on contact bird disease symptoms and feather viral load, for a model also including contact sex and shedder DPI but not including shedder vaccination status.** Full results in S1 Table.

| Contact bird response | Shedder viral load slope (SE) | Test statistic[6] | *p*-value |
|---|---|---|---|
| **Disease status**[1] | 3.83 (0.62) | 6.17 | <0.0001 *** |
| **Mortality**[2] | 0.93 (0.11) | 8.76 | <0.0001 *** |
| ***N* tissues with tumours**[3] | 0.34 (0.09) | 3.57 | <0.0005 *** |
| **Vagus nerve**[4] | 0.24 (0.16) | 1.51 | 0.13 |
| **Brachial nerve**[4] | 0.91 (0.17) | 5.4 | <0.0001 *** |
| **Sciatic nerve**[4] | 0.90 (0.16) | 5.57 | <0.0001 *** |
| **Feather viral load**[5] | 0.83 (0.08) | 10.33 | <0.0001 *** |

[1]Infected contacts (qPCR) only. Logistic regression. Coefficient = increase in log odds ratio for presence of contact disease symptoms with 1 unit increase in shedder FVL.

[2]Infected contacts (qPCR) only. Cox proportional hazards. Coefficient = increase in log hazard ratio of contact death at a given time point with 1 unit increase in shedder FVL.

[3]Diseased contacts (necropsy) only. Poisson GLM. Coefficient = increase in number of contact tissues containing tumours with 1 unit increase in shedder FVL.

[4]Diseased contacts (necropsy) only. Ordinal logistic regression. Coefficient = increase in proportional log odds of contact nerve enlargement ranking with 1 unit increase in shedder FVL.

[5]All contacts. Increase in contact FVL with 1 unit increase in shedder FVL.

[6]t statistic for linear regression, z statistic for all other models.

**Abbreviations:** DPI, days post-infection; FVL, feather viral load; GLM, generalized linear model; qPCR, quantitative polymerase chain reaction; SE, standard error

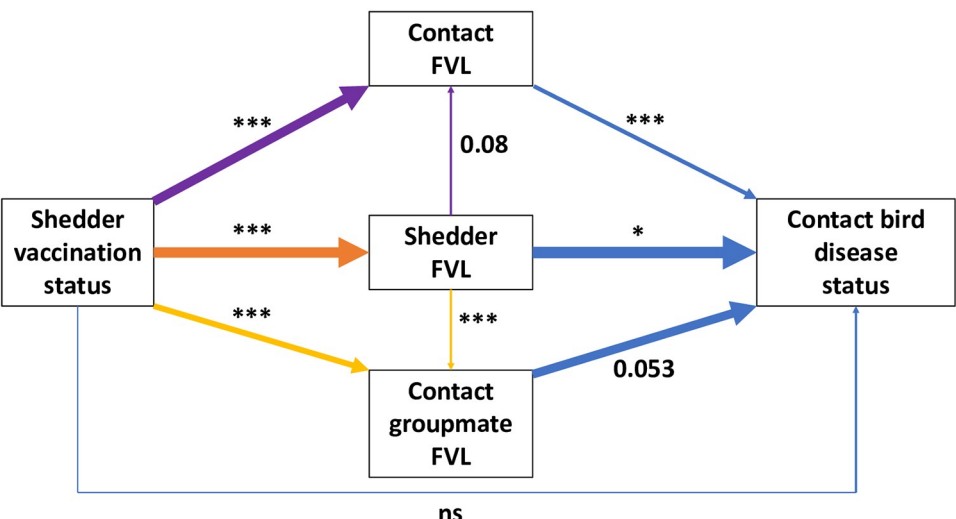

**Fig 5. Diagrammatic representation of the mediating effects of viral load on contact bird binary disease status.** Each arrow colour represents a different multiple regression analysis, with arrows pointing from predictors to response. Arrow thickness represents regression coefficient (all significant or marginal relationships were positive). The diagram shows that the effect of shedder FVL on contact FVL (see Table 2) fails to fully explain the shedder vaccination effect, which remains significant when both variables are included (purple arrows). The same is true for the cumulative FVL of all infected groupmates (yellow arrows). However, the 3 FVL predictor variables together completely remove the effect of shedder vaccination status (blue arrows) in explaining contact disease status (see Table 1), and this is true for all other contact disease variables (S1 Table). This implies that shedder vaccination effects on contacts are fully mediated by FVL of infected individuals in a contact group. Significance is indicated by asterisks: ***p < 0.001, **p < 0.01, *p < 0.05, ns = not significant. Marginally nonsignificant *p*-values are presented numerically. All results presented here are from models excluding the DPI by vaccination status interaction but including sex and DPI main effects (not shown, see S1 Table). Underlying data are in Edinburgh DataShare repository (https://doi.org/10.7488/ds/2725). DPI, days post-infection; FVL, feather viral load.

To examine whether the presence of even undetectably small quantities of vaccine virus in contact birds might affect the causal relationship between same-individual viral load and disease development, we carried out further multiple regression analyses, with contact bird pathogen FVL nested within shedder vaccination treatment. As expected in the absence of any extra effect of vaccine transmission, contact FVL remained significant within each vaccination treatment (except for vagus nerve, which was nonsignificant in results presented in Tables 1 and 2), with similar effect size estimates in each (S1 Table).

## Discussion

We used controlled experiments involving natural virus transmission to reveal that vaccination with a leaky vaccine, which only marginally reduces transmission, can significantly reduce post-transmission disease development and mortality among unvaccinated contact individuals. Our analysis indicates that this effect is mediated by a reduction in exposure dose experienced by susceptible individuals when exposed to vaccinated shedders, leading to lower pathogen load and concomitant reduced symptoms in contact birds. The primary objectives of vaccination of livestock with leaky vaccines are to improve animal welfare and to reduce production losses caused by disease symptom development. Our results show that even partial vaccination against MD can substantially reduce disease symptoms and mortality in the whole flock, leading to universally positive impacts on animal welfare and productivity, and these conclusions may extend to leaky vaccines used in other systems.

Modified live virus vaccines, such as HVT and other MD vaccines, have the potential to be transmitted and cause secondary vaccination [37–41], and this may partially explain our results. Unlike in previous studies, showing that HVT transmission only occurred from older birds vaccinated with higher doses [39–41], we found nonzero transmission of HVT from young birds vaccinated with a low dose. However, with HVT virus absent or below detectable levels in 91% of contacts of vaccinated shedders, HVT transmission would fail to explain the reduced contact bird MDV viral load and disease progression in our statistical analyses. We found that shedder FVL, as a proxy of exposure dose, did not always fully explain the shedder vaccination effects on contact birds. This may be because feather samples taken at the start of a 48-hour contact period provide an imperfect proxy for exposure dose. It may also be partly due to the presence of another factor, such as vaccine transmission, further influencing both contact bird viral load and disease progression and their associations. However, contact bird FVL strongly and equally explained disease progression in contacts of each of vaccinated and unvaccinated shedders, suggesting no additional factors influencing this relationship in the vaccinated treatment. Vaccine transmission nevertheless remains potentially important and should be addressed in future studies, for both MDV and other diseases treated with transmissible vaccines.

One of the key findings of this study was that shedder vaccination effects on MD symptom prevalence and subsequent mortality within each contact group were fully explained by the summed FVL of all infected group members, measured at a relatively early stage of the epidemic, prior to onset of contact-contact transmission. This would suggest that contacts exposed to vaccinated shedders experienced overall lower cumulative exposure dose, including from other infected contacts, over the course of 8 weeks, resulting in milder symptoms and lower mortality. This negative feedback on the environmental pathogen burden strongly advocates for the application of MD intervention strategies that reduce either within-host or environmental virus load, even if only moderately [47]. In general, depending on the relationship between exposure dose and subsequent within-host replication in any particular system,

targeting reduction in pathogen load in intervention strategies may have greater positive knock-on effects than currently assumed.

Increased disease severity with higher virus inoculation dose has been shown previously for MD in chickens [27,30] but not with natural transmission and not linked to interventions such as vaccination. The route of infection is known to alter the extent of infection and number of diseased tissues [48]; hence it is important to mimic the field situation closely in order to accurately predict the outcome of vaccination and other intervention strategies in this and other systems. It is also important to measure pathogen load specifically in the infectious tissues where possible, or to measure shedding directly, because tissues may differ in the strength of association between their pathogen load and infectiousness. Vaccine experiments routinely establish the protective effect of vaccination on infection or disease progression, and pathogen shedding, in vaccinated individuals themselves [49], and occasionally also examine onward transmission. The novelty and primary focus of this study was to determine the effects of vaccination of 'shedder' individuals on disease progression and infectiousness specifically in newly infected, unvaccinated, contact individuals. To date, little is known about these potential 'downstream' effects of vaccination, and the majority of epidemiological models that predict the consequences of vaccination on disease spread and pathogen evolution assume that these do not exist [3,50]. Our findings that vaccination affects downstream pathogen load and host survival and hence, potentially, onward pathogen transmission, in a dose-dependent manner may have profound consequences for such predictions. Particularly in systems in which an individual's infectiousness is strongly influenced by its pathogen load, existing estimates of the required vaccine coverage for achieving so-called herd immunity, i.e., for preventing disease spread within a population, may be upwardly biased. Based on current estimates, herd immunity is assumed to require high coverage when vaccines are leaky [18,51]. Complete vaccine coverage is not typical for all infectious diseases throughout the world, and, even where it is routine, vaccine administration can vary in quality. Furthermore, high variation in vaccine responsiveness may render a significant proportion of vaccinated animals effectively nonimmunized [52]. The results of our study suggest that partial vaccination or high prevalence of vaccine non-responders may impose less risk with respect to disease invasion and persistence than anticipated from existing theory [4,53,54]. Prediction of the coverage required for herd immunity would benefit from an understanding of the downstream effects of vaccination-induced changes in exposure dose and their effects on individuals' infectiousness and survival in any particular system. These insights would also be useful for the development of dynamic epidemiological models that incorporate dose-dependent transmission effects and the impact of interventions on these [19,24,55–57].

Vaccination with leaky vaccines has been implicated in the evolution of increased pathogen virulence, primarily because vaccination reduces host mortality without preventing pathogen transmission, allowing vaccinated infectious hosts more time to transmit virulent pathogen strains [3,5,50]. This reduced mortality with vaccination, when combined with higher shedding rate in more virulent strains (as has been found for MDV [5,58]), shifts the balance between transmission and host mortality in favour of higher virulence [58]. The best evidence for this vaccination effect on virulence evolution comes from studies of MDV [5]. Our results, showing that vaccination can also reduce mortality in unvaccinated contact individuals without strongly affecting transmission rates, indicate that virulence could evolve to higher levels than previously expected in a mixed flock. However, this expectation is specific to this particular combination of pathogen, vaccine and host strain, and depends on the relationship between vaccination and shedding rate. Previous studies have shown that the shedding rate of more virulent MDV strains is less affected by vaccination [5,59], further favouring their transmission, and this effect is thought to promote increased pathogen virulence in a rodent malaria system

[60,61]. However, in a system such as ours in which exposure dose is important, a smaller vaccine effect in reducing shedding of more virulent strains may not necessarily favour higher virulence in mixed flocks because of the resulting higher exposure dose and consequent increased downstream mortality. Further modelling is required to predict virulence evolution in mixed flocks in the presence of the downstream impacts presented here, and the differential effects of vaccination on viral shedding rate reported elsewhere. Furthermore, the lower pathogen load and slightly reduced rate of spread revealed here—the latter effect potentially being stronger in other systems—would lower the effective population size of the virus, therefore lowering the probability of establishment and fixation of new beneficial mutations such as those increasing virulence [62]. This may partially explain why years pass between reported increases in MDV virulence, despite 59 billion chickens being reared annually worldwide [34,63]. The results of this study therefore suggest that existing models of virulence evolution would benefit from incorporation of dose response effects on downstream disease severity and mortality such as those detected here.

Vaccination is often not the only available intervention technique, and others such as improved animal husbandry techniques or genomic selection of genetically more resistant individuals [64–66] may also be available and cost-effective. The mediating effect of pathogen load can be used as a link for comparison between the effects of different interventions on pathogen dynamics and disease severity. Chickens are known to vary genetically in their resistance to MD [64], and hence a next step in understanding the benefits of interventions in this system would be to compare vaccination effects with host genetic effects. Furthermore, more virulent virus strains tend to result in higher viral shedding rates and may differ in their response to vaccination [5]. Hence, future studies to test the validity of our findings for multiple MDV and vaccine strains are warranted.

With the increasing development of leaky vaccines for treatment of human as well as livestock infectious diseases [3], there is great benefit in improving prediction of their consequences for host welfare, pathogen dynamics, and virulence evolution. The currently neglected downstream, post-transmission effects we revealed in this study are likely to impact all of these important facets of infectious disease biology and hence disease management strategies. They therefore merit greater attention in future vaccine-related studies.

## Methods and materials

### Ethics statement

All bird experiments were approved by the Avian Disease and Oncology Laboratory IACUC, US National Poultry Research Center, United States Department of Agriculture, approval number: Avian Disease and Oncology Laboratory (ADOL) 2016–07. Our animal care and use protocol adhered to the Animal Welfare Act (AWA), United States Department of Agriculture (USDA), and Animal Plant Health Inspection Service (APHIS). All experiments were carried out in custom-made negative pressure Horsfall-Bauer isolators [67]. A scoring system was developed and approved by the ADOL IACUC for monitoring progression of MD and for determining humane endpoints (SOP #9). Humane endpoint criteria include body posture, neurological signs, eye closure, response to stimuli, and ability to eat and drink. Chickens experiencing clinical signs of MD were immediately humanely euthanized upon reaching the prescribed clinical sign score. The euthanasia method was carbon dioxide gas inhalation, based on AVMA Guidelines for the Euthanasia of Animals 2020.

## Transmission experiments

Experiments were carried out at USDA, ARS, USNPRC, ADOL, East Lansing, USA, during 2018. All experiments used $15I_5 \times 7_1$ white leghorn chickens, a F1 hybrid cross of MD-susceptible 15I5 males and $7_1$ females [43]. These maternal antibody-negative chickens were reared from a SPF breeding flock housed in isolators that have received no MD vaccination or exposure. The flock was negative for MDV antibodies and also for exogenous avian leukosis virus and reticuloendotheliosis virus, as established by routine surveillance testing.

The experiments involved 2 types of shedders, with shedder birds either vaccinated at hatch via intra-abdominal (IA) inoculation with 2,000 PFU of HVT (*Meleagrid alphaherpesvirus* 1) [33] or sham-vaccinated with PBS. Each shedder bird was then challenged with 500 PFU of virulent MDV (strain JM/102W) at 5 DPV (0 DPI). Each contact group of birds within each replicate consisted of 3 shedder birds of the same vaccination treatment (HVT or PBS) to be placed in contact with 15 unvaccinated, uninfected contacts (Fig 1). The 3 shedders were placed with the first group of 15 uninfected contacts at 13 DPI for 48 hours before being removed back to their isolator at 15 DPI. They were then placed with a second group of 15 contacts at 20 DPI until 22 DPI. Contact chicks were hatched weekly so that all contact birds were within 4 days of age when shedders were first introduced. There were 16 replicates consisting of paired lots of shedder birds (1 lot with 3 vaccinated shedders put into contact with 15 contacts at the 2 time points and the other with 3 sham-vaccinated shedders) and 4 further sham-vaccinated only replicates. These additional replicates were carried out because of early death of 2 sham-vaccinated shedders involved in the earlier replicates.

Shedders were then monitored until 8 weeks post-infection and contacts until 8 weeks post-contact and mortality (death or euthanasia) recorded. Necropsy was carried out at 8 weeks or upon death, whichever was the sooner, to determine the presence and severity of MD symptoms.

Blood (100 μl) and primary feather samples were taken from shedders at the start of each contact period (13 and 20 DPI) and from contacts at 14 DPC. Based on earlier experiments, 14 DPC was sufficient for build-up of virus in blood and feathers but early enough to avoid cross-contamination from other contact birds (S4 Text, S4 Fig). If HVT vaccine virus transmission occurred, 14 days would also be sufficient for HVT to replicate to close to its maximum viral load in the new host [36–38,68]. DNA samples isolated from feather pulp and PBLs were used for qPCR to determine virus load. Each measurement was taken from a unique sample.

DNA from each tissue type was isolated using the Puregene DNA isolation kit (Gentra System, Minneapolis, MN) followed by a multiplex PCR using methods as previously described for MDV [69] and HVT [70]. The TaqMan assay used FAM-TAM probes for virus gB and VIC-TAM probes for the cellular GAPDH. Results were reported as the ratio of virus gB copies per GAPDH copies, estimated using standard curves consisting of 10-fold serial dilutions of plasmids containing either virus gB or GAPDH. Amplifications were performed at Michigan State University, USA, using the ABI Quant Studio 7Flex BI 7500 (Carlsbad, California).

## Statistical analyses

At total of 42 of 1,080 contacts were removed from the data set prior to analysis because of chick mortality (death up to 7 days old), with some further filtering for data quality and death by other causes. Final sample sizes were 211 (shedder FVL as response), 1,005 (infected contacts only), 789 (diseased contacts only), and 1,023 (all contacts regardless of infection or disease status). The transmission experiments were analysed using various linear and generalized linear mixed models in R version 3.6.0 [71], depending on the type of the response variable (Table 3). Regression analyses followed the logic of process analysis [72] to assess the role of

**Table 3. Summary of modelled response variables.**

| Response variable | Description | Source | Coefficient interpretation | Statistical model | Data subset[6] |
|---|---|---|---|---|---|
| Disease status | Binary presence/absence of visible disease symptoms | Necropsy | Log odds | Logistic regression (GLM binomial errors)[1] | Infected |
| Mortality | Day of death/euthanasia or last day of study | Daily observations | Log proportional hazard ratio | Right-censored Cox proportional hazards[2] | Infected |
| N tissues with tumours | Number of tissues with visible tumours | Necropsy | Log relative risk | GLM Poisson errors[3] | Diseased |
| Nerve enlargement | Qualitative ranking of nerve enlargement (0–4) | Necropsy | Log proportional odds | Ordinal logistic regression[4] | Diseased |
| Viral load | $\log_{10}$(Ratio of virus to GAPDH quantity $+ 1 \times 10^{-5}$) | qPCR | Mean relative quantity | Ordinary linear regression[5] | All |

[1] R function glmer in lme4 package [74]; logit link.

[2] R function coxph in survival package [75].

[3] R function glmer in lme4 package; log link.

[4] R function clmm in ordinal package [76]; logit link.

[5] R function lmer in lme4 package; identity link.

[6] Infected = positive for virus in qPCR of one or both of feather and blood samples; Diseased = presence of visible disease symptoms (tumours and/or peripheral nerve enlargement) at necropsy; All = all contact individuals including uninfected.

**Abbreviations:** GAPDH, Glyceraldehyde 3-phosphate dehydrogenase gene; GLM, generalized linear model; qPCR, quantitative polymerase chain reaction

pathogen load in mediating shedder vaccination effects on contacts—details below. Nonmetric multidimensional scaling for Fig 2B was carried out in PC-ORD version 7.0 [73] statistical software. All statistical tests were two-sided. No adjustments were made for multiple comparisons.

First, we tested the direct treatment effect (shedder vaccination status) on the outcome variables (contact disease variables, Table 3). The model formulae also included as fixed effects contact bird sex and shedder DPI, and a vaccination status by DPI interaction, which was removed if nonsignificant. Replicate and contact group nested within replicate were included as random effects in all models except for the survival analysis, for which contact group and replicate were included as clustering variables. Each contact individual was treated as a data point. For this and all subsequent analyses, testing contact FVL as response involved all contact individuals, infected or uninfected (Table 3). Contact binary disease status and mortality analyses involved infected (from qPCR) contacts only, and disease severity variables (tumours and nerve enlargement) involved diseased (from necropsy) contacts only.

Second, we carried out a process analysis, for which we tested all intermediate steps in the following proposed causal chain (see Fig 5): We hypothesized that the impacts of shedder vaccination status on the various contact infection and disease variables were primarily mediated by the vaccine effect on shedder FVL. More specifically, we hypothesized that shedder vaccination directly reduces shedder FVL and consequently also the exposure dose of contacts. The resulting lower exposure dose may reduce the probability of becoming infected and/or may lead to lower ingestion dose and consequently also to lower viral load in infected contacts. Lower contact viral load reduces the probability in infected contacts of developing visible disease symptoms or dying within the 8-week experimental period and also reduces disease severity among individuals positive for symptoms at necropsy. Eight weeks is also sufficient time for infected contacts to become infectious themselves and for disease development to occur in contacts infected by other contacts. Hence it was necessary to also consider the FVL of infected group mates alongside shedder and contact FVL in the process analysis.

Transmission of HVT was nonzero but nevertheless too low in a subsample of 6 contact bird groups to explain the vaccination effect (see Results section) and was therefore not explicitly included in our process analysis. We began the process analysis by testing whether shedder FVL explained a similar amount of contact bird disease variation as shedder vaccination status, by replacing shedder vaccination status with shedder FVL in the model formula described in the first step above. We then tested to what extent shedder FVL was affected by vaccination and then to what extent contact FVL and the sum FVL of each contact bird's groupmates (hereafter denoted as groupmate FVL) were affected by vaccination and shedder FVL. Thus, for contact FVL and groupmate FVL as response variables, the model formulae were the same as described in the first step above, with the addition of sum of shedder FVL for each contact group as a fixed effect. Conversely, when shedder FVL was tested as a response variable, we used each individual shedder feather sample as a data point, and hence there were 2 data points per shedder individual (13 and 20 DPI). For this test, we used the same fixed effects model formula as described in the first step above, while replicate and shedder individual were included as random effects, the latter to account for repeated measures.

The values for contact FVL at 14 DPC were calculated as $\log_{10}(\text{contact FVL} + 1 \times 10^{-5})$ for each individual. The contact groupmate FVL variable was the sum of FVL at 14 DPC of all 15 contacts in a group, minus the value for the focal individual. This variable was also analysed as $\log_{10}(\text{groupmate FVL} + 1 \times 10^{-5})$. For shedder FVL as a predictor, we calculated $\log_{10}(\text{sum} (\text{shedder FVL} + 1 \times 10^{-5}))$ across the 3 shedders, from feather samples collected at the start of the contact period with each group of 15 contacts (13 and 20 DPI).

Third and finally, we tested whether shedder vaccination status exerted any effect on contact disease variables when controlling for mediating effects (shedder, contact, and contact groupmate FVL). If shedder vaccination status were to be rendered nonsignificant when tested alongside FVL variables, this would support the hypothesis that shedder vaccination impacts on contact disease were fully mediated by their effects on FVL. We first added shedder FVL alone to the basic model described in step 1, above, to test whether this variable was an effective bioindicator of shedder vaccination effects in secondary cases (infected contacts). We then further added contact and groupmate FVL to the model. Same-individual viral load is expected to be the strongest indicator of disease status, and so we expected shedder vaccination status and shedder and groupmate FVL to become nonsignificant in this model.

To examine whether the presence of even undetectably small quantities of vaccine virus in contact birds might affect the causal relationship between same-individual viral load and disease development, we carried out further multiple regression analyses with contact FVL nested within shedder vaccination treatment. For each response variable, we used a mixed-effects model with the same random effects as described above, and fixed effect predictors shedder vaccination status, shedder DPI, and contact bird sex alongside the nested contact FVL predictor.

## Supporting information

**S1 Text. Pilot study design summary.**
(DOCX)

**S2 Text. Determination of the duration of contact between shedder and contact birds required for successful virus transmission.**
(DOCX)

**S3 Text. Establishing the onset and duration of the infectious period of vaccinated and sham-vaccinated MDV-infected shedder birds.** MDV, Marek disease virus.
(DOCX)

**S4 Text. Determination of appropriate sampling time post-contact for measuring FVL in contact birds.** FVL, feather viral load.
(DOCX)

**S1 Fig. Effects of exposure duration on contact bird MD.** For each tested contact duration, the proportion of line $15I_5 \times 7_1$ $F_1$ contact birds positive for MD symptoms at necropsy, 8 weeks post-contact with inoculated unvaccinated 'MD-resistant' line 6 (blue line) or 'MD-susceptible' line 7 (red line) shedder birds. MD, Marek disease.
(TIF)

**S2 Fig. FVL over time for shedder birds.** Vaccinated (blue) and sham-vaccinated (red) shedders, with maximum likelihood broken stick regression lines indicating lower viral load and a later breakpoint in viral load over time for vaccinated shedders. Open circles = replicate 1, crosses = replicate 2. The shaded area encompasses the set of shedder DPI during which contact occurred between shedders and contact birds. DPI, days post-infection; FVL, feather viral load.
(TIF)

**S3 Fig. Impact of shedder vaccination status and DPI on contact bird infection, disease symptoms, and mortality.** Contacts positive for virus in qPCR from samples taken at 14 DPC were classified as infected. 'Diseased' individuals showed visible symptoms (peripheral nerve enlargement and/or tumours) at necropsy, 8 weeks post-contact or upon death. 'Dead' individuals died because of MD prior to the end of the 8-week experimental period. HVT = contacts exposed to vaccinated shedders; PBS = contacts exposed to sham-vaccinated shedders. The 2 replicates were pooled for this figure. DPC, days post-contact; DPI, days post-infection; HVT, herpesvirus of turkeys; MD, Marek disease; qPCR, quantitative polymerase chain reaction.
(TIF)

**S4 Fig. Effect of number of days post-contact on contact bird FVL.** Histogram of contact bird FVL from qPCR at 7 (red bars) and 14 (blue bars) days post-contact with unvaccinated infectious shedders (2 replicates and all 4 shedder chicken lines combined). A value of −5 indicates negative for MDV; i.e., values were below the level of detection by standard qPCR. FVL, feather viral load; MDV, Marek disease virus; qPCR, quantitative polymerase chain reaction.
(TIF)

**S1 Table.**
(XLSX)

# Acknowledgments

The authors would like to thank the many farm staff and animal caretakers that provided daily care to the birds. Mention of trade names or commercial products in this publication is solely for the purpose of providing specific information and does not imply recommendation or endorsement by the US Department of Agriculture.

# Author Contributions

**Conceptualization:** Hans H. Cheng, Margo Chase-Topping, Osvaldo Anacleto, John R. Dunn, Andrea Doeschl-Wilson.

**Data curation:** Hans H. Cheng, Jody K. Mays, John R. Dunn.

**Formal analysis:** Richard I. Bailey, Andrea Doeschl-Wilson.

**Funding acquisition:** Hans H. Cheng, John R. Dunn, Andrea Doeschl-Wilson.

**Investigation:** Richard I. Bailey, Jody K. Mays, Osvaldo Anacleto, John R. Dunn, Andrea Doeschl-Wilson.

**Methodology:** Hans H. Cheng, Jody K. Mays, Osvaldo Anacleto, John R. Dunn, Andrea Doeschl-Wilson.

**Project administration:** Hans H. Cheng, John R. Dunn, Andrea Doeschl-Wilson.

**Resources:** Hans H. Cheng, Jody K. Mays, John R. Dunn.

**Supervision:** Hans H. Cheng, John R. Dunn, Andrea Doeschl-Wilson.

**Validation:** John R. Dunn.

**Visualization:** Richard I. Bailey, Margo Chase-Topping.

**Writing – original draft:** Richard I. Bailey.

**Writing – review & editing:** Richard I. Bailey, Margo Chase-Topping, John R. Dunn, Andrea Doeschl-Wilson.

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
