## [Editor Report · Decision Letter 0]

3 Sep 2019

Dear Dr Bailey, 

Thank you for submitting your manuscript entitled "Transmission from vaccinated hosts can cause dose-dependent reduction in pathogen virulence" for consideration as a Research Article by PLOS Biology.

Your manuscript has now been evaluated by the PLOS Biology editorial staff as well as by an academic editor with relevant expertise and I am writing to let you know that we would like to send your submission out for external peer review.

*Please be aware that, due to the voluntary nature of our reviewers and academic editors, manuscripts may be subject to delays during the holiday season. Thank you for your patience.*

Please re-submit your manuscript within two working days, i.e. by Sep 05 2019 11:59PM.

Kind regards,

Lauren A Richardson, Ph.D

Senior Editor

PLOS Biology

---

## [Decision Letter · Decision Letter 1]

30 Sep 2019

Dear Dr Bailey,

Thank you very much for submitting your manuscript "Transmission from vaccinated hosts can cause dose-dependent reduction in pathogen virulence" for consideration as a Research Article at PLOS Biology. Your manuscript has been evaluated by the PLOS Biology editors, an Academic Editor with relevant expertise, and by several independent reviewers.

As you will read, the reviewers appreciated many aspects of your work. However, they do raise some concerns that will need to be addressed in a revision. Of particular importance, two of the reviewers believe that since the HVT vaccine is also shed into the environment from vaccinated animals, this must be accounted for in your studies. Additionally, the Academic Editor recommends that you extend your discussion of how this work relates to the previous studies on virulence evolution in Marek’s disease.

In light of the reviews (below), we will not be able to accept the current version of the manuscript, but we would welcome resubmission of a much-revised version that takes into account the reviewers' comments. We cannot make any decision about publication until we have seen the revised manuscript and your response to the reviewers' comments. Your revised manuscript is also likely to be sent for further evaluation by the reviewers.

Your revisions should address the specific points made by each reviewer. Please submit a file detailing your responses to the editorial requests and a point-by-point response to all of the reviewers' comments that indicates the changes you have made to the manuscript. In addition to a clean copy of the manuscript, please upload a 'track-changes' version of your manuscript that specifies the edits made. This should be uploaded as a "Related" file type. You should also cite any additional relevant literature that has been published since the original submission and mention any additional citations in your response. 

Before you revise your manuscript, please review the following PLOS policy and formatting requirements checklist PDF: http://journals.plos.org/plosbiology/s/file?id=9411/plos-biology-formatting-checklist.pdf. It is helpful if you format your revision according to our requirements - should your paper subsequently be accepted, this will save time at the acceptance stage.

Please note that as a condition of publication PLOS' data policy (http://journals.plos.org/plosbiology/s/data-availability) requires that you make available all data used to draw the conclusions arrived at in your manuscript. If you have not already done so, you must include any data used in your manuscript either in appropriate repositories, within the body of the manuscript, or as supporting information (N.B. this includes any numerical values that were used to generate graphs, histograms etc.). For an example see here: http://www.plosbiology.org/article/info%3Adoi%2F10.1371%2Fjournal.pbio.1001908#s5.

For manuscripts submitted on or after 1st July 2019, we require the original, uncropped and minimally adjusted images supporting all blot and gel results reported in an article's figures or Supporting Information files. We will require these files before a manuscript can be accepted so please prepare them now, if you have not already uploaded them. Please carefully read our guidelines for how to prepare and upload this data: https://journals.plos.org/plosbiology/s/figures#loc-blot-and-gel-reporting-requirements.

Upon resubmission, the editors will assess your revision and if the editors and Academic Editor feel that the revised manuscript remains appropriate for the journal, we will send the manuscript for re-review. We aim to consult the same Academic Editor and reviewers for revised manuscripts but may consult others if needed.

We expect to receive your revised manuscript within two months. Please email us (plosbiology@plos.org) to discuss this if you have any questions or concerns, or would like to request an extension. At this stage, your manuscript remains formally under active consideration at our journal; please notify us by email if you do not wish to submit a revision and instead wish to pursue publication elsewhere, so that we may end consideration of the manuscript at PLOS Biology.

When you are ready to submit a revised version of your manuscript, please go to https://www.editorialmanager.com/pbiology/ and log in as an Author. Click the link labelled 'Submissions Needing Revision' where you will find your submission record. 

Sincerely,

Lauren A Richardson, Ph.D

Senior Editor

PLOS Biology

Reviews

Reviewer #1: 

In this paper, the authors report results of experiments testing for downstream impacts of vaccination by evaluating disease severity in chickens infected by chickens that received the leaky Marek’s vaccine relative to those infected by chickens that received a sham control. Chickens that were infected by vaccinated birds experienced reduced symptoms, and this effect was shown to result from reduced feather virus load in vaccinated birds. The authors argue that these results demonstrate leaky vaccines may be more protective than generally assumed. I enjoyed reading this clear and well-written paper and believe it will stimulate further research into the downstream impacts of vaccination. 

Specific comments:

1. The title should be changed to: “Pathogen transmission from vaccinated hosts…” as the original title strongly suggests the paper focuses on transmissible vaccines and is misleading. 

2. The figures appear blurry, at least in my copy.

3. In figure 2B. After staring at this figure for a while and repeatedly reading the legend, I do not understand what message it is intended to convey. Also, what explains the odd clustering of points into discrete bands? Is each odd band/cluster a replicate?

---------------

Reviewer #2: 

Both MDV1 and HVT (vaccine strain) are shed from the infected or vaccinated birds, respectively. It is known that the levels of HVT shedding are increased by 38 folds when the HVT vaccinated birds are infected with MDV1 (Islam A, Walkden-Brown SW. J Gen Virol 2007).

It is conceivable that HVT (as well as MDV1) is transmitted from these birds to the unvaccinated contact birds. Transmission of HVT to the unvaccinated contact birds may inhibit disease progression (unwanted vaccination). Therefore, both lower MDV1 dose as well as possibility of HVT transmission to the unvaccinated birds may be involved in the lack of clinical signs of the disease in spite of virus replication.

No attempt has been made to examine the role of HVT transmission to the unvaccinated contact groups, and analyze its effects on the positive consequences in controlling the spread ad symptoms of the disease.

---------------

Reviewer #3:

General Comment: 

In the submitted manuscript, the authors investigated the effect of leaky vaccines on the transmission of an avian herpesvirus to unvaccinated contact chickens. They claim that the reduced replication of Marek’s disease virus (MDV) in vaccinated animals has a protective effect on the contact chickens. While this information would be very exciting to a broad audience, their experiments do not provide sufficient convincing evidence to support this conclusion (yet). For example, the authors completely disregard the fact that the HVT vaccine is also shed into the environment and could therefore provide some protection to the contact animals. This and other aspects are outlined in the major and minor points below.

Major Points:

1) The main message of the manuscript is that the use of leaky vaccines reduces the severity and frequency of clinical symptoms of Marek’s disease virus (MDV) in contact chickens. Unfortunately, the author completely disregard the fact that this leaky HVT vaccine is also shed into the environment from vaccinated animals, and also upon challenge with MDV [1, 2]. Therefore, the contact animals likely acquired HVT that in turn would provide some protection against MDV. In poultry industry, animals are vaccinated just hours before they get exposed to MDV in a contaminated environment, confirming that a concurrent vaccination can efficiently provide protection. To exclude this likely scenario that undermines their conclusions, the authors should perform the following experiments. 

a. Since the contact animals could have acquired HVT, the authors should test the blood (and feathers) of the contact animals for HVT genome copies by qPCR. 

b. In addition, they should test if the dust contains HVT genome copies, as even HVT antigens released into the environment could have some protective effect in the contact animals and explain the outcome of their animals experiment. 

c. The authors could also use a spread deficient HVT vaccine to proof that spead of HVT is not responsible for the observed phenotype. 

2) Along those lines. It would be worth testing if contact animals of HVT vaccinated animals (mock infected) also show reduced clinical symptoms when they are challenged with MDV. 

3) The authors claim that the reduced shedding of MDV from HVT vaccinated animals contributes to the reduced clinical symptoms in the contact animals. However, very few plaque forming units are sufficient to induced disease and was shown by various laboratories. The authors previously demonstrated that cohousing infected and uninfected animals for 4 hours is sufficient for transmission, indicating that not massive amounts of virus are needed to infect contact animals. They cohoused animals for 48 hours, which would be more than enough despite the lower shedding levels. The authors should consider and discuss these aspects in their manuscript. 

Minor Points:

1) Virus remains infectious – not “viable”

2) Line 128: Shedder experiments are common in the field. So stating that they used a novel setup is overstating it. 

3) Why did the authors not use a more current vaccine like SB-1 or CVI988? And why did they use a low virulent virus instead of MD5 or vv+ strains for their experiments? The authors should discuss these points. 

1. Islam, A.; Walkden-Brown, S. W., Quantitative profiling of the shedding rate of the three Marek's disease virus (MDV) serotypes reveals that challenge with virulent MDV markedly increases shedding of vaccinal viruses. J Gen Virol 2007, 88, (Pt 8), 2121-8.

2. Islam, A. F. M. F.; Groves, P. J.; Underwood, G. J.; Walkden-Brown, S. W., Dynamics of Marek's disease virus and herpesvirus of turkey shedding in feather dander of broiler chickens. In Poultry Research Foundation: Sydney, 2005; pp 105-108.

---

## [Decision Letter · Decision Letter 2]

18 Dec 2019

Dear Dr Bailey,

Thank you for submitting your revised Research Article entitled "Pathogen transmission from vaccinated hosts can cause dose-dependent reduction in virulence" for publication in PLOS Biology. I have now obtained advice from one of the original reviewers and have discussed their comments with the Academic Editor. 

Based on the reviews, we will probably accept this manuscript for publication, assuming that you will modify the manuscript to address the remaining points raised by the reviewer. Please also make sure to address the data and other policy-related requests noted at the end of this email.

We expect to receive your revised manuscript within three weeks. Your revisions should address the specific points made by each reviewer. In addition to the remaining revisions and before we will be able to formally accept your manuscript and consider it "in press", we also need to ensure that your article conforms to our guidelines. A member of our team will be in touch shortly with a set of requests. As we can't proceed until these requirements are met, your swift response will help prevent delays to publication.

*Copyediting*

*Published Peer Review History*

*Early Version*

*Submitting Your Revision*

Sincerely,

Lauren A Richardson, Ph.D

Senior Editor

PLOS Biology

ETHICS STATEMENT:

The Ethics Statements in the submission form and Methods section of your manuscript should match verbatim. Please ensure that any changes are made to both versions.

-- Please include the full name of the IACUC/ethics committee that reviewed and approved the animal care and use protocol/permit/project license. Please also include an approval number.

-- Please include the specific national or international regulations/guidelines to which your animal care and use protocol adhered. Please note that institutional or accreditation organization guidelines (such as AAALAC) do not meet this requirement.

DATA POLICY:

*Thank you for providing so much of your data already. Please make it more clear to the reader which data are used in each figure. 

Review

Reviewer #3: 

General Comment: 

In this manuscript, the authors investigated the effect of leaky vaccines on the transmission of an avian herpesvirus to unvaccinated contact chickens. Their data suggests that the reduced replication of Marek's disease virus (MDV) in vaccinated animals has a protective effect on the contact chickens. In this resubmission of the manuscript PBIOLOGY-D-19-02539, the authors addressed most of the reviewer's comments and improved the manuscript, which is now suitable for publication.

Minor Points:

1) Line 364: it should be "91% of contact birds" and not 90%.

---

## [Editor Report · Decision Letter 3]

30 Jan 2020

Dear Dr Bailey,

On behalf of my colleagues and the Academic Editor, David S. Schneider, I am pleased to inform you that we will be delighted to publish your Research Article in PLOS Biology. 

Early Version

PRESS 

Kind regards,

Vita Usova 

Publication Assistant, 

PLOS Biology

on behalf of

Lauren Richardson,

Senior Editor

PLOS Biology